# The Crucial Role of Hereditary Cancer Panel Testing in Unaffected Individuals with a Strong Family History of Cancer: A Retrospective Study of a Cohort of 103 Healthy Subjects

**DOI:** 10.3390/cancers16132327

**Published:** 2024-06-25

**Authors:** Lucrezia Pilenzi, Federico Anaclerio, Anastasia Dell’Elice, Maria Minelli, Roberta Giansante, Michela Cicirelli, Nicola Tinari, Antonino Grassadonia, Andrea Pantalone, Simona Grossi, Nicole Canale, Annalisa Bruno, Giuseppe Calabrese, Patrizia Ballerini, Liborio Stuppia, Ivana Antonucci

**Affiliations:** 1Center for Advanced Studies and Technology (CAST), “G. d’Annunzio” University of Chieti-Pescara, 66100 Chieti, Italy; lucrezia.pilenzi@studenti.unich.it (L.P.); anastasia.dellelice@studenti.unich.it (A.D.); mariaminelli60@gmail.com (M.M.); roberta.giansante@phd.unich.it (R.G.); michela.cicirelli@studenti.unich.it (M.C.); ntinari@unich.it (N.T.); antonino.grassadonia@unich.it (A.G.); a.bruno@unich.it (A.B.); patrizia.ballerini@unich.it (P.B.); stuppia@unich.it (L.S.); i.antonucci@unich.it (I.A.); 2Department of Medical Genetics, “G. d’Annunzio” University of Chieti-Pescara, 66100 Chieti, Italy; 3Department of Medical, Oral and Biotechnological Sciences, “G. d’Annunzio” University of Chieti-Pescara, 66100 Chieti, Italy; 4Department of Innovative Technologies in Medicine and Dentistry, “G. d’Annunzio” University of Chieti-Pescara, 66100 Chieti, Italy; 5Orthopaedic and Traumatology Department, “G. d’Annunzio” University of Chieti-Pescara, 66100 Chieti, Italy; andrea.pantalone@unich.it; 6U.O.C. Chirurgia Generale ad Indirizzo Senologico, Eusoma Breast Center ASL2 Abruzzo, 66026 Ortona, Italy; sgrossi@unich.it (S.G.); canalenicole@gmail.com (N.C.); 7Department of Hematology, Pescara Hospital, 66100 Pescara, Italy; giuseppe.calabrese@unich.it; 8Department of Psychological, Health and Territorial Sciences, “G. d’Annunzio” University of Chieti-Pescara, 66100 Chieti, Italy

**Keywords:** hereditary cancer, unaffected family member, NGS multigene panel

## Abstract

**Simple Summary:**

The purpose of this study is to emphasize the importance of genetic testing for healthy individuals with a strong family history of hereditary malignancies. A total of 103 healthy subjects with at least two relatives with cancer were enrolled. By NGS analysis of 27 genes, 5% were found to carry a pathogenic variant in a hereditary cancer susceptibility gene. In the era of personalized medicine, genetic testing of healthy subjects in the absence of a living affected collateral is crucially important for early diagnosis, clinical surveillance and surgical choice.

**Abstract:**

Hereditary cancer syndromes caused by germline mutations account for 5–10% of all cancers. The finding of a genetic mutation could have far-reaching consequences for pharmaceutical therapy, personalized prevention strategies, and cascade testing. According to the National Comprehensive Cancer Network’s (NCCN) and the Italian Association of Medical Oncology (AIOM) guidelines, unaffected family members should be tested only if the affected one is unavailable. This article explores whether germline genetic testing may be offered to high-risk families for hereditary cancer even if a living affected relative is missing. A retrospective study was carried out on 103 healthy subjects tested from 2017 to 2023. We enrolled all subjects with at least two first- or second-degree relatives affected by breast, ovarian, pancreatic, gastric, prostate, or colorectal cancer. All subjects were tested by Next Generation Sequencing (NGS) multi-gene panel of 27 cancer-associated genes. In the study population, 5 (about 5%) pathogenic/likely pathogenic variants (PVs/LPVs) were found, while 40 (42%) had a Variant of Uncertain Significance (VUS). This study highlights the importance of genetic testing for individuals with a strong family history of hereditary malignancies. This approach would allow women who tested positive to receive tailored treatment and prevention strategies based on their personal mutation status.

## 1. Introduction

Genetic testing for hereditary cancer risk is a strategy increasingly used in risk management and treatment planning. Indeed, it is well established that the identification of individuals with deleterious mutations in cancer susceptibility genes has clinical implications for affected people and their families [1]. Family investigations reveal an increased risk for multiple cancer types among first-degree relatives (parents, siblings, and children) and second-degree relatives (grandparents, aunts or uncles, grandchildren, nieces, or nephews) of affected individuals [2]. This may be due to pathogenic variants in parental germline cells. In the mid-1990s, *BRCA1* and *BRCA2* gene variants, exposing individuals to a higher risk of breast and ovarian cancer, were discovered [3]. Furthermore, by genetic linkage analysis, DNA sequencing, and positional cloning techniques, additional genes whose mutations are associated with moderate and low risk were identified (Figure 1) [4]. Genetic testing is generally indicated when there is a personal or family history consistent with an inherited predisposition to cancer [5]. According to the American Cancer Society’s guidelines, genetic testing should be recommended for people (i) with a strong family history of certain types of cancer; (ii) diagnosed with cancer when other factors suggest a likely inherited predisposition to cancer (remarkable familiarity, early-onset cancer, or uncommon cancer, i.e., male breast cancer); and (iii) relatives of a person known to carry an inherited gene mutation increasing their cancer risk [6]. When a patient has a causative mutation, it is advisable to include first-degree relatives in the analysis, as each family member has a 50% probability of carrying the same mutation [7]. In particular, genetic testing may be recommended for cancer-unaffected individuals with collateral ovarian tumors or early-onset breast cancer, bilateral disease, male breast cancer, numerous primary tumors, or additional malignancies linked to a probable hereditary condition, which are typically autosomal dominant [8]. Healthy carriers can benefit from risk management strategies, such as screening, chemoprevention, and risk-reducing prophylactic surgery for breast and ovarian cancer [7]. National and international guidelines recommend that unaffected family members should be tested only when the affected one is unavailable, emphasizing that testing affected relatives is more informative than testing healthy members [9]. Examining unaffected individuals without examining affected family members can pose significant challenges. Assessing multiple family members might be necessary, as the absence of a pathogenic variant in one unaffected relative does not preclude its presence in other family members. It is essential to analyze both the maternal and paternal sides of the family to identify familial cancer patterns accurately [10]. It is critical to address serious limitations in interpreting test results, as most are negative or non-informative due to the presence of unknown significance variations. Thus, it is important to emphasize that if a pathogenetic mutation is not inherited, the risk of developing it is similar to that of the general population [7]. In this scenario, genetic counseling is crucial in explaining the limited significance of “uninformative” results, and management should focus on other risk factors rather than on test results [11]. Few studies have explored genetic testing in unaffected subjects, with the percentage of positive carriers being less than 5%. For example, Trottier et al. found that 2.8% of unaffected women with a family history of breast or ovarian cancer had a pathogenic variant in *BRCA1/2* [12]. However, in recent years, multigene panel testing has emerged as a crucial approach for detecting clinically significant variants in individuals at high risk for cancer predisposition genes [13]. The present study involved around 100 unaffected individuals, selected only based on their familiar cancer history in the absence of a positive familiar member. Furthermore, we focused on the potential impact of a Next Generation Sequencing (NGS)-based multi-gene panel of 27 genes, including *BRCA1/2* genes, with the aim to understand whether expanding the analysis to a larger panel of genes may result in a percentage of healthy subjects with cancer-predisposing gene variants higher than that reported in previous studies [13,14,15].

## 2. Materials and Methods

### 2.1. Study Population

One-hundred and three healthy subjects (95 women and 8 men), who during pre-counseling tests reported a significant familiar history for hereditary cancers, were retrospectively retrieved among those consecutively referred to the Medical Genetic Service of the University “G. d’Annunzio” of Chieti-Pescara–Center of Advanced Studies and Technologies (CAST) from 2017 to 2023. In these families, anyone underwent genetic testing. All subjects with at least two first- or second-degree relatives with breast, ovarian, pancreatic, gastric, prostate, or colorectal cancer were enrolled in the study. During genetic counseling, which was provided by a multidisciplinary team of geneticists, psychologists, and physicians, personal and familiar histories were acquired. All subjects were informed about the significance of the genetic test, the possible implications of detecting the gene variant related to increased cancer risk, and available prevention strategies. All subjects signed an informed consent form. No cancer risk tools were used during the counseling. The results of the analysis and their implications were explained during the post-test counseling.

### 2.2. Genomic DNA Extraction

Buccal swabs or blood samples were collected from all subjects during the pre-counseling test. Genomic DNA was extracted using the MagPurix instrument and the Forensic DNA Extraction Kit (Zinexts Life Science Corp., Taipei, Taiwan-CatZP01001) or Blood DNA Extraction Kit 200 (Zinexts Life Science Corp., Taipei, Taiwan-CatZP02001), according to the manufacturer’s protocol.

### 2.3. Next-Generation Sequencing (NGS)

NGS analysis was carried out with a Thermo-fisher Oncomine custom panel developed in our laboratory, including 27 genes (Table 1). NGS was performed through the Ion Torrent S5 system (Thermo Fisher Scientific, Waltham, MA, USA) after automatic library preparation using Ion Chef (Thermo Fisher Scientific, Waltham, MA, USA). The Ion Chef system allows fragmentation and adapter ligation onto the PCR products, which is called clonal amplification. After quantification of DNA libraries with the Real-Time Step One PCR System (Thermo Fisher Scientific, Waltham, MA, USA), the prepared samples of ion sphere particles (ISP) were loaded onto an Ion 530TM chip with the Ion Chef (Thermo Fisher Scientific, Waltham, MA, USA). Sequencing was performed using the Ion S5TM sequencing reagents (Thermo Fisher Scientific, Waltham, MA, USA). The Torrent Suite 5.14.0 platform and specific plugins were used for NGS data analysis. The uniformity of base coverage was over 99% in all batches, and base coverage was over 20× in all target regions.

### 2.4. Sanger Sequencing

Specific pathogenic variants identified in each subject via NGS were confirmed via Sanger sequencing. All DNA samples were amplified via polymerase chain reaction (PCR) performed in a 30 μL reaction volume containing 22.25 μL of H_2_O, 3 μL of 10X PCR buffer, 2.1 μL of MgCl_2_ solution 25 mM, 0.5 μL of dNTPs 10 mM, 0.15 μL of AmpliTaq Gold polymerase, 1 μL of DNA, and 0.5 μL of Forward and 0.5 μL of Reverse primers. All primers were designed using NCBI designing tools (https://www.ncbi.nlm.nih.gov/tools/primer-blast/, accessed on 12 October 2023). The list of primers used to confirm the analysis is reported in Table 2. Amplification was performed via a SimpliAmpTM thermal cycler (ThermoFisher, Applied Biosystem, Foster City, CA, USA). A FastGene Gel/PCR Extraction kit (Nippon Genetics Europe, Düren, Germany) was utilized for the purification of PCR products, according to the manufacturer’s protocol. The amplification products were submitted to the direct sequencing procedure using the BigDye Term v3.1 CycleSeq Kit (Life Technologies, Monza, Italy), followed by automatic sequencing analysis. All sequences were purified via the “NucleoSEQColumns” purification kit (Macherey-Nagel Colonia, Düren, Germany) and analyzed in forward and reverse directions on a SeqstudioGenetic Analyzer (ThermoFisher, Applied Biosystem, Foster City, CA, USA).

### 2.5. Genetic Variant Classification

According to the guidelines of the Evidence-based Network for the Interpretation of Germline Mutant Alleles (ENIGMA) (https://enigmaconsortium.org/ accessed on 23 April 2024), genetic variants were classified into five classes: benign (C1), likely benign (C2), variant of uncertain significance (VUS, C3), likely pathogenic (C4), and pathogenic (C5). In the present study, we focused on the pathogenic variants that can be used for cancer prevention. The variants were referred to according to the nomenclature recommendations of the Human Genome Variation Society (https://www.hgvs.org). The clinical significance of the genetic variants found in this study was evaluated according to ClinVar (https://www.ncbi.nlm.nih.gov/clinvar), Varsome (https://varsome.com), Franklin Genoox (https://franklin.genoox.com) and, for some other susceptibility genes, according to LOVD-InSIGHT (https://www.insight-group.org/variants/databases/).

## 3. Results

The mean age of the 103 healthy subjects was 49 years (range 28–65). A panel of 27 cancer susceptibility genes was examined. The prevalence of pathogenic variants was 5%. In particular, among five discovered pathogenic variants, two were detected in *BRCA1* and *BRCA2*, and one each in *CHEK2, POLE*, and *MUTYH* (Table 3). Of these, four females were positive, and only one man had a PV in the *POLE* gene. All these pathogenic variants were in the heterozygous state. Out of 103 unaffected individuals tested, 36 (35%) had a VUS in 18 different genes, including *ATM, BARD1, BRCA1, BRCA2, PALB2*, and *CHEK2*, for a total of 40 variants classified as C3. The remaining 62 subjects (60%) showed neither a deleterious variant nor VUS.

### 3.1. Genes Variants Linked to the Homologous Recombination (HR) and Related Family History

The study revealed two crucial aspects in the familial history of the tested healthy subjects. In many cases, several members of the family were affected by the same or by a different type of cancer, while in others, at least one family member developed cancer before the age of 50 years. Starting from the genetic analysis of subject 1, we found the c.4914dupA pathogenic mutation located in exon 10 of the *BRCA2* gene that causes a translational frameshift with a predicted alternate stop codon (V1639fs). The family history of this subject revealed three relatives affected: the mother with gastric cancer at the age of 65, the maternal grandfather with colorectal cancer at 77, and the maternal aunt with breast cancer at 37 (Figure 2), all deceased at the time of the pre-test counseling.

Subject 2 showed the c.1427C>T LPV in the *CHEK2* gene, resulting in a damaging effect with reduced or absence of kinase activity and DNA damage response [16]. The familial history of this case revealed a grandmother with ovarian cancer at 69 and a maternal aunt with ovarian cancer at 51. In other cases, we found the same type of cancer in two family members.

During the pre-counseling test, subject 3 reported a sister and a paternal aunt deceased from ovarian cancer at 54 and 58 years, respectively. In this subject, genetic testing evidenced a pathogenic mutation in the *BRCA1* gene, the c.1953dup located in coding exon 9, resulting from a duplication of Guanine at nucleotide position 1953, that leads to a translational frameshift with a predicted alternate stop codon (K652fs) [17].

Case number 4 presented a pathogenic mutation in the *POLE* gene, the c.778C>T, which creates a premature translational stop signal and is expected to result in an absent or disrupted protein product with loss of function (R260*) [18]. In this family, there was a significant presence of tumors, and many of them were different from each other, with a few family members having the same type of cancer. On the maternal side, the mother had ovarian cancer at 50, and a first-degree uncle had throat cancer at 55. On the paternal side, the father had throat cancer at 70 and, subsequently, colorectal cancer. Other tumors registered in additional family members were colorectal cancer in a paternal aunt and in a cousin, both at 55, bladder cancer in a first-degree uncle at the age of 55, throat cancer in a paternal cousin at 45 who was also a smoker, lung cancer in a paternal uncle at 85, and ultimately, another cousin deceased at 45 from brain cancer (Figure 3).

### 3.2. Genes Variants Linked to the Base Excision Repair (BER) and Related Family History

The father of subject 5 was diagnosed with colorectal cancer at the age of 41, a first-degree aunt had breast cancer at 47, and a first-degree uncle had colorectal cancer at 45. The PV found, in this case, was the c.1187G>A in the MUTYH gene, located in coding exon 13 and causing the substitution of a Glycine with Aspartate in codon position 396. This alteration is frequently reported as a founder mutation in multiple populations. M. Nielsen et al. have shown that this missense variant changes the function of the MUTYH protein [19].

## 4. Discussion

In recent years, several genes associated with hereditary cancer syndromes have been identified, and at least 2% of presumably healthy individuals carry highly penetrating pathogenic gene variants predisposing them to cancer [20]. Individuals with hereditary cancer syndromes have a higher risk of developing multiple primary cancers during their lifetime or may develop cancer at a younger age.

Early mutation detection and prevention are key aspects of managing hereditary cancer risks [21], thus supporting the implementation of targeted screening and prevention strategies in this population. In this regard, genetic testing may play a crucial role. However, according to international and national guidelines, when a familial predisposition is suspected, genetic testing should be preliminarily performed on a family member who has already developed the disease (index case). Alternatively, when the index case is unwilling to perform the test, it is possible to offer the genetic test to a healthy relative who has a high probability of mutation (>10%) during the entire lifetime [22]. The present study evaluated, for the first time, the presence of pathogenic mutations in healthy subjects with a strong family history of hereditary cancers in the absence of affected living relatives. As an inclusion criterion, the presence of at least two first- or second-degree relatives with breast, ovarian, pancreatic, gastric, prostate, or colorectal cancer was used. In this population of healthy subjects, a 5% prevalence of pathogenic mutations in genes correlated with high or moderate risk of developing cancer was found. This finding highlights the need to extend genetic testing to healthy individuals with suggestive familiarity, even in the absence of an index case. In particular, the family trees of two probands (Figure 2 and Figure 3) were reported, in which a pathogenic mutation in *BRCA2* c.4914dupA and *POLE* c.778C>T, respectively, was detected. In both cases, distinct forms of cancers were found in the generations portrayed. In particular, the family of subject 4 with a *POLE* mutation exhibits a significant cancer history from both maternal and paternal lines, thus emphasizing the importance of studying the entire family history to select subjects eligible for testing. A study by Magrin et al. reported the importance of genetic tests for germline PVs/LPVs also in the *POLE* gene for people belonging to a cancer family where hereditary cancers have already been present [22]. Moreover, the French Genetic and Cancer Group-Unicancer recommended including not only *POLE* but also *POLD1* in multi-gene panel genetic tests to evaluate the predisposition to hereditary cancer of the digestive tract. These results suggest that a change in current eligibility criteria for genetic testing in healthy subjects could be evaluated. Specifically, we propose that testing should be conducted in healthy subjects with the following family history features: (I) at least two first-degree relatives with one of the following cancers: breast, ovarian, pancreatic, gastric, prostate, and colorectal; (II) at least one first-, second-, or third-degree relative with an early-onset tumor (<45 years old); (III) at least two relatives affected by the same type of cancer. Furthermore, we propose to perform the test before the age of 55 since in clinical practice, instrumental preventive screening starts at the age of 25 or 10 years before the age of cancer, beginning in the youngest affected family member [23]. The NCCN guidelines for clinical practice in oncology offer specific recommendations and surveillance programs tailored to the type of mutated gene detected in healthy individuals, such as imaging modalities, frequency of evaluation, and risk-reducing surgery. This proactive approach seeks to diagnose cancer in its earliest, most treatable stages or to entirely prevent its development. Furthermore, genetic testing and the discovery of a mutation associated with an elevated risk of cancers is critical not only for enrolling the proband in tailored surveillance programs but also for healthy collaterals. Once the mutation in the family has been identified, testing should be extended to all members. This could allow the assessment of their likely carrier status and their enrollment in surveillance and therapy programs. A recent study by Di Rado et al. underscores the importance of cascade testing in at-risk relatives of probands with PV/LPV in one of 27 cancer susceptibility genes [7]. Segregation among relatives strengthens the association between identified variants and cancer predisposition. Carriers of pathogenic mutations can benefit from appropriate risk management and preventative strategies due to an inherited increased risk of breast, ovarian, prostate, melanoma, and pancreatic cancers. Cascade testing could significantly increase the identification of pathogenic variant carriers, as 70% of probands may inform their family members, and 20% may receive genetic testing, potentially increasing the 5% of pathogenetic variants found in the present work. It is important to emphasize how the use of the multigenic panel, including 27 genes associated with hereditary cancers, allowed us to increase the detection rate of unaffected individuals with mutations in genes beyond *BRCA1* and *BRCA2*. In addition, we remark that the cancer risk assessment should not be based only on the presence or absence of a low penetrant pathogenic variant or risk factor variant. In fact, in the study of Stolarova et al. [24], where they evaluated 460 VUS of the *CHEK2* gene, they assumed that the clinical needs clear discrimination between the PVs and non-PVs because they could change the clinical management of carriers. This study demonstrated that variants of *CHEK2* only had a moderate association with BC risk, but none with other tumors analyzed. So, in the absence of multi-gene panel analysis, a considerable percentage of mutations would have been lost.

The use of a multi-gene panel significantly reduces both the time and cost of analysis from a cost–benefit perspective. Avoiding stringent criteria to select healthy patients with familiarity to submit NGS testing allows a larger cohort and reduces NGS testing costs. In addition, the early detection of a pathogenic mutation and the inclusion of healthy individuals in surveillance programs may significantly reduce cancer-related healthcare costs. So far, our experience with hereditary cancer panel testing in unaffected individuals has been very encouraging. All five families with identified mutations allowed women to make decisions about surgical risk reduction based on their personal mutation state. Interestingly, our research found that women are more likely to request genetic testing, possibly due to their long-standing knowledge of the benefits of being identified as BRCA carriers, while men are more likely to participate for their daughters. Additionally, unaffected individuals under 40 were found to be more likely to request pre-test counseling appointments.

In this view, genetic counseling importantly contributes to the analysis process by providing people with information on their risk, available preventive measures, chemotherapy protocols, and other treatment options, such as preventive surgery. Moreover, genetic counseling represents a strategic tool for identifying the criteria that suggest the presence of a mutation in the proband’s family tree and the decision to consider him or her eligible for testing. On these bases, the role of pre-test counseling in explaining the limited significance of “uninformative” results appears crucial. Geneticists must clarify how mutations in moderate penetrance genes vary from *BRCA1/2* mutations in terms of risks. Certainly, the involvement of multiple family members in the decision regarding a genetic test can potentially lead to tensions and disagreements. In the future, it will be crucial to overcome social and cultural barriers hindering effective communication between families. Additionally, the psychological impact of predictive genetic testing should not be overlooked. Understanding mutation status can cause psychological distress due to the high lifetime risks of cancer development, so psychological support is recommended during pre- and post-genetic counseling [11,25,26]. Challenges and future directions should focus on supporting intrafamilial communication and improving communication processes between professionals and at-risk relatives. In addition, radiological screening, surgeons, gynecologists, and “omics” approaches should play a crucial role in identifying high-risk individuals for hereditary cancer predisposition syndromes prevention. [27] It is possible to outline a hypothetical scenario based on the extension of *BRCA* genetic testing to healthy women in the general population. Population screenings would also reduce the overall costs associated with managing these hereditary syndromes, offsetting the additional costs resulting from increased genetic testing. From a practical clinical point of view, there is a need to develop strategies to improve test uptake by unaffected individuals. In this context, it has been recently proposed that the use of a web-based tool may result in higher quality collection of cancer family history compared to clinician collection, thus improving the percentage of participants completing genetic counseling and testing [28].

## 5. Conclusions

In conclusion, the present study is the first to implement hereditary cancer gene panel testing for high-risk families, providing individuals with mutations a more informed choice for individual decisions on surveillance or risk-reducing surgery. This approach is crucial in the era of personalised cancer prevention and early detection. Despite its limitations, our findings offer preliminary insights that can serve as a baseline for future research.

## Figures and Tables

**Figure 1 cancers-16-02327-f001:**
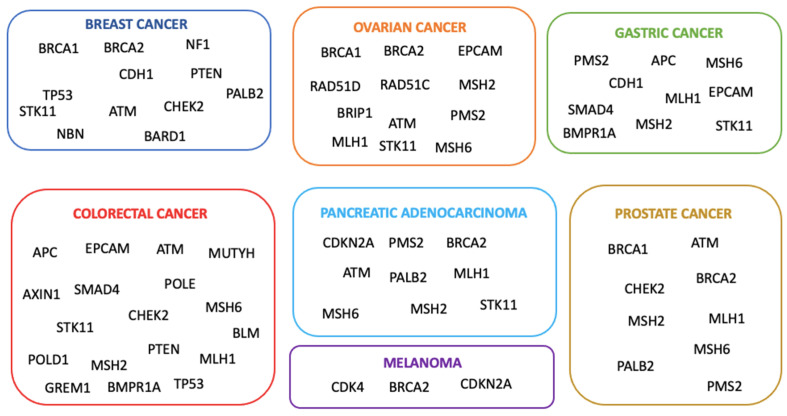
The major hereditary cancers and their correlated genes of predisposition.

**Figure 2 cancers-16-02327-f002:**
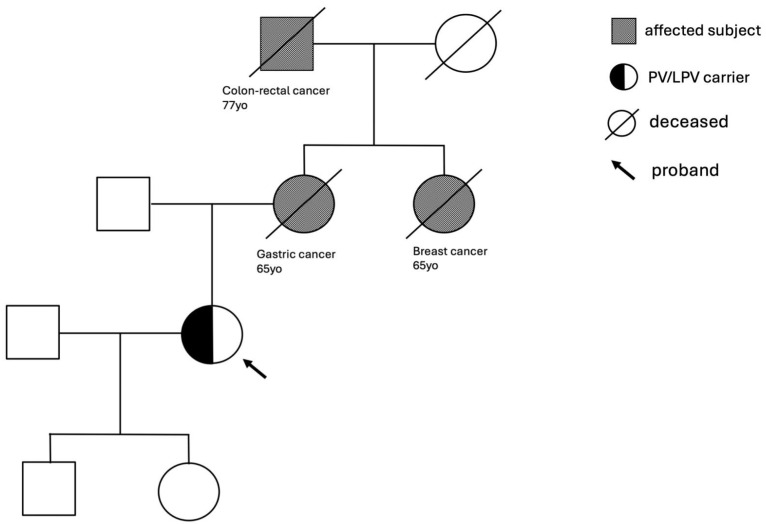
Family tree of subject 1 with *BRCA2* c.4914dupA pathogenic mutation.

**Figure 3 cancers-16-02327-f003:**
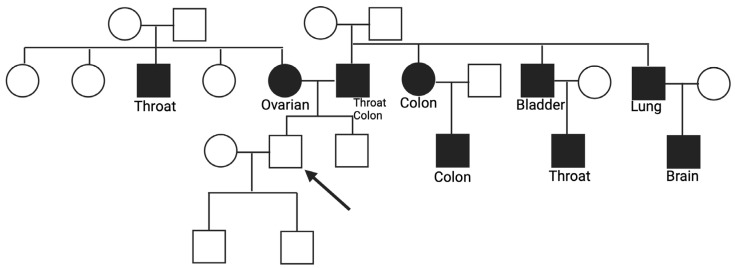
Familiar history of subject 4 with a PV c.778C>T in the *POLE* gene.

**Table 1 cancers-16-02327-t001:** The 27 genes included in the NGS multi-gene panel.

	Multi-Gene Panel	
*APC*	*ATM*	*BARD1*
*BRCA1*	*BRCA2*	*BRIP1*
*CDK4*	*CDK12*	*CDKN2A*
*CDH1*	*CHEK2*	*EPCAM*
*MLH1*	*MSH2*	*MSH6*
*MUTYH*	*NBN*	*NF1*
*PALB2*	*POLE*	*POLE*
*POLD1*	*PTEN*	*RAD51C*
*RAD51D*	*SMAD4*	*TP53*

**Table 2 cancers-16-02327-t002:** List of primers designed to confirm the PV/LPV found in the 5 healthy subjects.

NAME	EXON	SEQUENCE
BRCA2_EX11F	Exon 11	attgagatcacagctgcccc
BRCA2_EX11R	Exon 11	tgaagtctgactcacagaagttt
CHEK2_EX13F	Exon 13	atgtggatgtgagtcagccag
CHEK2_EX13R	Exon 13	atcagctccttaagcccagacta
BRCA1_EX10F	Exon 10	ttggtcagctttctgtaatcg
BRCA1_EX10R	Exon 10	ccataccacgacatttgaca
POLE_EX8F	Exon 8	gtcgctgctcacatgaattt
POLE_EX8R	Exon 8	atttgggggaaaagcagcaa
MUTYH_13F	Exon 13	agggcagtggcatgagtaac
MUTYH_13R	Exon 13	gggtcaaggggttcaaatag

**Table 3 cancers-16-02327-t003:** All PVs/LPVs found in healthy carriers (* identifies stop codon).

CASE ID	GENE	OMIM	REFSEQ	CODING	PROTEIN
Subject 1	*BRCA2*	164757	NM_000059.3	c.4914dupA	V1639fs
Subject 2	*CHEK2*	604373	NM_007194.4	c.1427C>T	T476M
Subject 3	*BRCA1*	113705	NM_007294.4	c.1953dup	K652fs
Subject 4	*POLE*	174762	NM_006231.3	c.778C>T	R260*
Subject 5	*MUTYH*	608456	NM_001128425.2	c.1187G>A	G396D

## Data Availability

Data are available upon request.

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
