# Peer review of "The Crucial Role of Hereditary Cancer Panel Testing in Unaffected Individuals with a Strong Family History of Cancer: A Retrospective Study of a Cohort of 103 Healthy Subjects"

_cancers, 2024, doi:10.3390/cancers16132327_

Round 1
Reviewer 1 Report
Comments and Suggestions for Authors This article raises the crucial question of the benefits of genetic counselling for patients with a variety of cancers. It is well organised and covers many different tumour types, using not only anamnestic data but also extensive genetic testing. I would recommend more clearly presenting the specific genes that should be tested for each hereditary cancer, as it is impossible to test all potential genes and not all genes contribute equally to hereditary cancers. Therefore, this information should be presented more clearly.Author Response
Dear Reviewer,
We thank you for taking the time to review the manuscript and we appreciate it your contribution about.
Please find the detailed responses below and the corresponding revisions highlighted in the re-submitted files. Kind regards Federico Anaclerio

Reviewer 2 Report
Comments and Suggestions for Authors
Comments for authors
The article gives an interesting information on what was the result of germline multi-gene panel testing of healthy individuals with family history of cancer in their cohort of patients.
Here are some suggestions for authors.
Under the section
2. Materials and Methods.
Study population
Was there any assessment of cancer risk, or the risk of being a carrier of BRCA mutation based on family history and other info with cancer risk prediction programs?
NGS.
Are all exons of genes listed in Table 1 sequenced?, are sequenced also the exon/intron boundaries? Are splice mutation detected by this test?
3. Results.
Table 3. The zygosity of variants should be added to variants /or allelic frequencies.
Results line 187-191: The variant detected in CHEK2 c.1427C>T is probably a low penetrant pathogenic variant, or established risk factor and, if in heterozygous state, is not necessary associated with ovarian cancer. Pathogenic variants are associated more with breast cancer. The cancer risk assessment in case of low penetrant pathogenic variant or risk factor, should not be based only on the presence or absence of this variant. It would be suggested to discuss this in discussion.
Ref: Lenka Stolarova et al, ENIGMA CHEK2gether Project: A Comprehensive Study Identifies Functionally Impaired CHEK2 Germline Missense Variants Associated with Increased Breast Cancer Risk, Clin Cancer Res 2023;29:3037–50, doi: 10.1158/1078-0432.CCR-23-0212.
Results - line 190: What did author mean with “In other cases, we found the same type of cancer in two family members” ?
Results lines 197-209:
One of the pathogenic variants, that authors had listed in results is truncating variant c.778C>T R260*in exonuclease domain of POLE gene.
Loss-of-function variants of POLE do not cause the cancer predisposition syndrome PPAP (polymerase proofreading associated Polyposis syndrome); however, they may predispose to autosomal recessive or dominant congenital disorders (FILS syndrome, very rare recessive Mendelian disorder characterized by facial dysmorphism, immunodeficiency, livedo, short stature, and variable skin manifestations, is caused by POLE pathogenic variants disrupting (truncating) protein). Biallelic POLE pathogenic variants have also been associated with another rare Mendelian syndrome, IMAGE-I (MIM# 618336), characterized by intrauterine growth retardation, metaphyseal dysplasia, adrenal hypoplasia congenita, genital anomalies, immunodeficiency, and diffuse large B-cell lymphoma.
Reference: https:// doi.org/ 10. 1186/ s13073-​023-​01234-y
POLE pathogenic variants associated with cancer predisposition are missense variant located in the exonuclease domain.
Authors are citing the article, that describes a case of a girl with growth retardation, microcephaly, developmental delay, dysmorphic features, poikiloderma, immune deficiency with pancytopenia, and myelodysplasia, being homozygous for POLE splice variant. – which again is not associated with hereditary cancer predisposition syndrome.
The POLE truncating variant does not explain the presence of tumors in the family where it was detected, and it should be considered, if it counts as finding associated with hereditary cancer syndrome. Since the article is talking about hereditary cancer syndrome, this should be discussed by authors in Discussion.
Throat cancer major risk factors include infection with the human papillomavirus (HPV) as well as tobacco and alcohol use.- They are usually not associated with hereditary cancer syndromes
Is the POLE variant present in homozygous state? In the results the heterozygous or homozygous state should be stated for all variants.
Results - line 199: What did author mean with “In this family, there was a significant and heterogeneous presence of tumors.”?
Comment on MUTYH PV. PV in MUTYH is connected to increate risk of cancer only if present as homozygous or biallelic.
Discussion:
Some points are mentioned upstream near results-comments.
Lines 250-252: It is not very clear what authors try to say with:
»Furthermore, we propose to perform the test before the age of 55 rather than 25, since in clinical practice instrumental preventive screening starts at age of 25 or 10 years before the age of cancer beginning in the youngest affected family member [22]. »
References:
Authors should check the reference 4
Author Response
Dear Reviewer,
We thank you for taking the time to review the manuscript and we appreciate it your contribution about.
Please find the detailed responses below and the corresponding revisions highlighted in the re-submitted files. Kind regards Federico Anaclerio
